# Staging dementia based on caregiver reported patient symptoms: Implications from a latent class analysis

**Qi Yuan**[1]*, **Tee Hng Tan**[1], **Peizhi Wang**[1], **Fiona Devi**[1], **Hui Lin Ong**[1], **Edimansyah Abdin**[1], **Magadi Harish**[2], **Richard Goveas**[2], **Li Ling Ng**[3], **Siow Ann Chong**[1], **Mythily Subramaniam**[1,4]

1 Research Division, Institute of Mental Health, Singapore, 2 Department of Geriatric Psychiatry, Institute of Mental Health, Singapore, 3 Department of Psychological Medicine, Changi General Hospital, Singapore, 4 Neuroscience & Mental Health, Lee Kong Chian School of Medicine, Singapore

* Qi_YUAN@imh.com.sg

## Abstract

### Background

Tailoring interventions to the needs of caregivers is an important feature of successful caregiver support programs. To improve cost-effectiveness, group tailoring based on the stage of dementia could be a good alternative. However, existing staging strategies mostly depend on trained professionals.

### Objective

This study aims to stage dementia based on caregiver reported symptoms of persons with dementia.

### Methods

Latent class analysis was used. The classes derived were then mapped with disease duration to define the stages. Logistic regression with receiver operating characteristic curve was used to generate the optimal cut-offs.

### Results

Latent class analysis suggested a 4-class solution, these four classes were named as early (25.9%), mild (25.2%), moderate (16.7%) and severe stage (32.3%). The stages based on the cut-offs generated achieved an overall accuracy of 90.8% compared to stages derived from latent class analysis.

### Conclusion

The current study confirmed that caregiver reported patient symptoms could be used to classify persons with dementia into different stages. The new staging strategy is a good complement of existing dementia clinical assessment tools in terms of better supporting informal caregivers.

**Data Availability Statement:** All individual data from this study resides with Office of Research, Institute of Mental Health. Data is not available for online access, however readers who wish to gain

access to the data can write to the Clinical Research Committee, Institute of Mental Health/ Woodbridge Hospital Secretariat at IMHRESEARCH@imh.com.sg. Access can be granted subject to the Institutional Review Board (IRB) and the research collaborative agreement guidelines. This is a requirement mandated for this research study by our IRB and funders.

**Funding:** The study is funded by the Singapore Ministry of Health's National Medical Research Council under the Center Grant Programme (Grant No.: NMRC/CG/004/2013) and the Institute of Mental Health Bridging Fund (CRC ref No.: 545-2016). The funders had no role in study design, data collection and analysis, decision to publish or preparation of the manuscript.

**Competing interests:** The authors have declared that no competing interests exist.

**Abbreviations:** ADL, Activities of Daily Living Scale; AIC, the Akaike Information Criteria; BIC, the Bayesian Information Criteria; cAIC, the consistent Akaike Information Criteria; CES-D, the Centre for Epidemiological Study Scale; IADL, Instrumental Activities of Daily Living Scale; LCA, latent class analysis; MBP, memory and behavior problems; PWD, persons with dementia; RMBPC, the Revised Memory and Behaviour Problems Checklist; ROC, receiver operating characteristic.

# Introduction

Dementia is an umbrella term that includes multiple diseases affecting memory, cognitive abilities, and behavior that interfere significantly with the individuals' ability to maintain their daily functioning [1]. In 2016, there were around 47 million persons with dementia (PWD) globally [2]. Alzheimer's Disease International suggested that for people aged 60 and above, the incidence of dementia doubles with every 6.3-year increase in age [3]. Consequent to the population aging [3], there will be 2 billion people aged 60 years and above globally in 2050 [4]; the number of PWD is also expected to increase, to an estimated number of 65.7 million in 2030 and 115.4 million in 2050 [5]. An increasing trend was also noticed in the financial costs associated with dementia. According to a 2017 systematic review, the total economic cost of dementia had escalated from US$279.6 billion in 2000 to US$948 billion in 2016, with an annual growth rate of 15.94% [6]; As a result, dementia has become one of the most important public health priorities [4].

PWD usually require a high level of care, and oftentimes they are taken care of by family members or other informal caregivers [7–9]. Informal caregiving refers to the activities and experiences involved in providing help and assistance to relatives or friends who are unable to care for themselves [10]. This evokes positive feelings among caregivers, such as a sense of satisfaction from observing improvements in the care recipient, fulfilling of parental obligation, or simply from the act of caring for a loved one [11]. However, in many cases, caregivers taking care of PWD tend to view these experiences as enduringly uncertain, stressful, and frustrating [12, 13]. Such adverse experiences might lead to high caregiving burden [8], or depressive symptoms [14]. Poor health of the caregivers can affect the patient as well, as it has been shown to result in lower quality of care and earlier institutionalization [7]. Therefore, it is imperative to support informal caregivers and to help them through the caregiving process.

Tailoring interventions to the needs of caregivers of PWD is an important component of effective caregiver support programs [7]. Mittelman and colleagues [15] found that individually tailored counseling sessions specifically targeting individual caregiver needs successfully reduced caregiver depressive symptoms over two years. This is consistent with the findings from a previous systematic review [16], suggesting that individually tailored behavior management therapy successfully reduces caregiver distress and burden both in the short and long term. However, individually-tailored interventions usually cost much more than group-based programs, resulting in low cost-effectiveness. To address this issue, one solution could be to adopt a medium level of customization—identifying specific subgroups of people who share common features or challenges and then tailoring interventions based on their specific demands. From this perspective, stages of dementia could be a good reference point, and it is also widely used in fields such as tailored e-health interventions for caregivers of PWD [17].

Several scales are used to determine the stage of dementia, such as the Clinical Dementia Rating scale [18] and the Mini-Mental State Examination [19]. However, these scales often require input from trained professionals [20]. In order to better support caregivers of PWD, assessments based on patient symptoms will be more operational i.e., caregivers can easily identify the stage of the PWD they are caring for and then search for specific caregiving tips. There have been efforts on staging dementia based on symptom profiles of PWD tracked through a web-based tool [21]; however, this tool has a full list of 60 symptoms, which is quite complex. The staging process relies on artificial intelligence and it is not easy for the caregivers to navigate this process. Thus, there is a need to stage dementia more practically and in a simpler way.

The current study aims to 1) use latent class analysis (LCA) to identify the latent subtypes of PWD based on caregiver reported patient symptoms, and to map these subtypes with illness

duration to stage dementia; 2) estimate the association between the subtypes and depressive symptoms among the caregivers; and 3) calculate the optimal cut-off values based on the assessment tools used to develop screening cutoff scores which can be used to classify different stages of PWD.

## Materials and methods

### Participants and procedures

Primary informal caregivers of PWD were recruited through convenience sampling from the outpatient clinics of a tertiary mental health hospital and a general hospital in Singapore. A local Voluntary Welfare Organization serving caregivers helped to advertise the study by displaying the study flyers so that potential participants could contact the study team. Participants had to meet the following eligibility criteria: 1) Singapore citizens or permanent residents, 2) aged 21 years or above, 3) able to read, write, and speak in English/Chinese/Malay, 4) self -identified as the primary informal caregiver of PWD (defined as family member or friend most involved in providing care or ensuring provision of care to the patients). Caregivers were excluded if they had difficulty understanding the consent (as a result we could not obtain informed consent) and if they failed to visit the PWD at least once on a weekly basis.

Data were collected from Jan 2017 to Dec 2018. Potential participants were mainly approached by the study team members or referred by our collaborating clinicians (two geriatric psychiatrists from the tertiary mental health hospital, and one from the general hospital). Eligible caregivers who were interested in the study were followed up by the study team member through an interviewer-administered questionnaire for data collection. On average, the length of the interview (on the scales used in the current study) took around 20 minutes. A total of 433 primary informal caregivers of PWD were approached, among which 282 agreed to participate, representing a response rate of 65%.

The study was approved by the National Healthcare Group Domain Specific Review Board in Singapore (reference number: 2016/00921). The study was conducted in accordance with the precepts of the Declaration of Helsinki. All participants signed the written informed consent prior to their enrolment, and appropriate measures were taken to ensure confidentiality and data privacy.

### Measurements

In the current study, symptoms of PWD were characterized by functional status, memory, and behavior problems. Functional status was measured by the widely used Activities of Daily Living Scale (ADL) [22] and the Instrumental Activities of Daily Living Scale (IADL) [23]. Both scales have been validated in Singapore as a measure of disability [24–26]. The ADL has 6 items, covering patient disability in six basic self-care activities (i.e., bathing, dressing, toileting, transfer, continence and feeding). The IADL includes 8 items, and it measured eight other self-care activities (i.e., ability to use a telephone, shopping, food preparation, housekeeping, laundry, mode of transportation, responsibility for own medication, and ability to handle finances). During the interview, caregivers were required to report whether the PWD received any assistance in any of these activities at the time of recruitment. Dependence of PWD was defined as receiving assistance from caregivers, and this included those who might be able to undertake the activity but refused to do so. The number of ADL and IADL dependencies were summed up separately to indicate the self-care impairment scores for the PWD. The memory and behavior problems of PWD were measured by the Revised Memory and Behaviour Problems Checklist (RMBPC) [27]. The RMBPC has 24 items in total, and it has three components, namely memory (7 items), disruptive behavior (8 items), and depression (9 items) [27].

Following a previous study [28], only the 15 items on memory and disruptive behaviour were used in the current study. Caregivers were asked to report whether these problems occurred during the week prior to their participation in the current study. Therefore, items under the ADL (independent vs. needs assistance), the IADL (independent vs. needs assistance), and the RMBPC (yes vs. no) were all binary variables.

Depressive symptoms among caregivers were measured by the 20-item Centre for Epidemiological Study Scale (CES-D) [29]. This scale was shown to have good validity and utility in detecting depression among family caregivers of PWD in Singapore [30]. Individuals were asked to rate how often they experienced depressive symptoms over the past week on a 4-point Likert scale (ranging from 0–3, with '0' representing 'rarely or none of the time and '3' representing 'most or all of the time). This scale has been widely used in epidemiological studies [31], and its total score ranges from 0 to 60, with higher scores indicating more severe depressive symptoms. A CES-D score of 16 or higher indicates at risk for clinical depression [32].

Socio-demographic information including caregiver's age, gender, ethnicity, marital status, education level, employment status and personal monthly income were collected. Caregiving related variables including caregiver's relationship to the PWD, living arrangement, having a domestic helper or not, and PWD's duration of being diagnosed with dementia were also collected.

## Data analysis

Four steps of statistical analyses were used to ultimately develop the screening criteria of dementia stages based on caregiver reported patient symptoms (i.e. ADL, IADL, and RMBPC):

1. latent class analysis was conducted using PROC LCA in SAS 9.3 [33] with regards to the scales to measure caregiver reported patient symptoms (i.e. ADL, IADL and RMBPC). The Akaike Information Criteria (AIC) [34], the Bayesian Information Criteria (BIC) [35], the consistent Akaike Information Criteria (cAIC) [36] and the interpretability of competing solutions [37] were considered while selecting the model with the optimal number of latent classes. Low information criteria indicate better fitting. Similar to previous studies [38, 39], interpretability was considered when information criteria contradicted; and a model with latent prevalence less than or equal to 10% was considered as limited clinical relevance in the current study.

2. the latent subtypes were mapped based on PWD's duration of diagnosis to identify if there is a clear trend of latent classes by time—stages in other words.

3. logistic regression analysis was used to explore the association between latent subtypes of symptoms of PWD and the depressive symptoms among the caregivers (depressive = 1 if CES-D >= 16).

4. logistic regression with receiver operating characteristic (ROC) curve was conducted to identify the optimal cut-off values based on the assessment tools used. Similar analytical strategies have been used in other studies [40–42]. The Youden's J statistic (sensitivity + specificity -1) [43] was used in the cut-off selection.

## Results

The mean age of the 282 participants was 55.7 years (standard deviation = 11.8), with three quarters of them being female (75.2%), and a majority being Chinese (83.0%). Around half of the primary caregivers were daughters of the PWD (55.3%), followed by sons (17.0%) and

spousal caregivers (15.3%). Most of them were living with the PWD (70.2%), and more than half of them had a domestic helper (57.1%). The descriptive statistics are shown in Table 1.

Comparisons of fit indices across solutions showed the AIC reached its lowest value in the seven-class solution, whereas the BIC and cAIC favored the four-class solution. Please refer to Table 2 for the model fit indices. In this case, solutions with classes from four to seven were all considered. However, the six-class and seven-class solutions both had a low prevalence for one of the classes (5.0% in six-class model; and 5.6% in seven-class model); and in the five-class model, one class was not readily distinguished from another class. As a result, also considering the interpretability, the four-class model was selected. Fig 1 plots the conditional probability of

**Table 1.  Descriptive statistics of the study participants (n = 282).**

|  | Frequency | Percentage |
|---|---|---|
| **Gender** | | |
| Male | 70 | 24.8 |
| Female | 212 | 75.2 |
| **Ethnicity** | | |
| Chinese | 234 | 83.0 |
| Malay | 29 | 10.3 |
| Indian & others | 19 | 6.7 |
| **Education level** | | |
| Secondary or below (including N/O level) | 120 | 42.6 |
| A level, polytechnic and other diploma | 73 | 25.9 |
| Degree or above | 89 | 31.6 |
| **Marital status** | | |
| Never married | 79 | 28.0 |
| Ever married | 203 | 72.0 |
| **Employment status** | | |
| Unemployed/retired/housewife | 121 | 42.9 |
| Employed | 161 | 57.1 |
| **Monthly Income** | | |
| <SGD2,000 (USD 1,477) | 56 | 19.9 |
| SGD2,000 (USD 1,477)–SGD5,999 (USD 4,431) | 78 | 27.7 |
| SGD6,000 (USD 4,432) or above | 42 | 14.9 |
| Not applicable | 106 | 37.6 |
| **Relationship to the PWD** | | |
| Spouse | 43 | 15.3 |
| Son | 48 | 17.0 |
| Daughter | 156 | 55.3 |
| Others | 35 | 12.4 |
| **Living Arrangement** | | |
| Together with the PWD | 198 | 70.2 |
| Separately from the PWD | 84 | 29.8 |
| **Have a domestic helper** | | |
| Yes | 161 | 57.1 |
| No | 121 | 42.9 |
|  | Mean | Standard deviation |
| **Age** | 55.7 | 11.8 |

Note: SGD—Singapore dollars; PWD—persons with dementia

**Table 2. Comparisons of model fit indices for fitted LCA models.**

| Number of classes | Log-likelihood | AIC | BIC | cAIC | Entropy |
|---|---|---|---|---|---|
| 4 | -3716.59 | 4494.69 | **4928.08** | **5047.08** | 0.89 |
| 5 | -3654.86 | 4431.24 | 4973.88 | 5122.88 | 0.89 |
| 6 | -3615.89 | 4413.28 | 5065.18 | 5244.18 | 0.92 |
| 7 | -3577.82 | **4397.14** | 5158.3 | 5367.3 | 0.94 |

Note: AIC—the Akaike Information Criteria; BIC—the Bayesian Information Criteria; cAIC—the consistent Akaike Information Criteria

class based on PWD symptoms. After mapping the classes with duration of illness, these four mutually exclusive classes were sequenced following the order of class 4 –class 2 –class 1 –class 3. Class 4 (25.9%) was characterized as low dependence on ADL, low dependence on IADL and low memory and behavior problems (MBP); class 2 (25.2%)—low dependence on ADL, low dependence on IADL and high MBP; class 1 (16.7%)—high dependence on ADL, high dependence on IADL and high MBP; class 3 (32.3%)—high dependence on ADL, high dependence on IADL, low MBP. This is consistent with the disease progression of dementia, thus class 4 is named as 'early stage', class 2 –'mild stage', class 1 –'moderate stage', and class 3 –'severe stage'. Caregivers who were taking care of PWD in the moderate (55.3%) and mild stages (45.1%) showed much higher prevalence of potential depression compared to those providing care to PWD in the early (31.5%) or severe stages (26.4%). Please refer to Table 3 for the details.

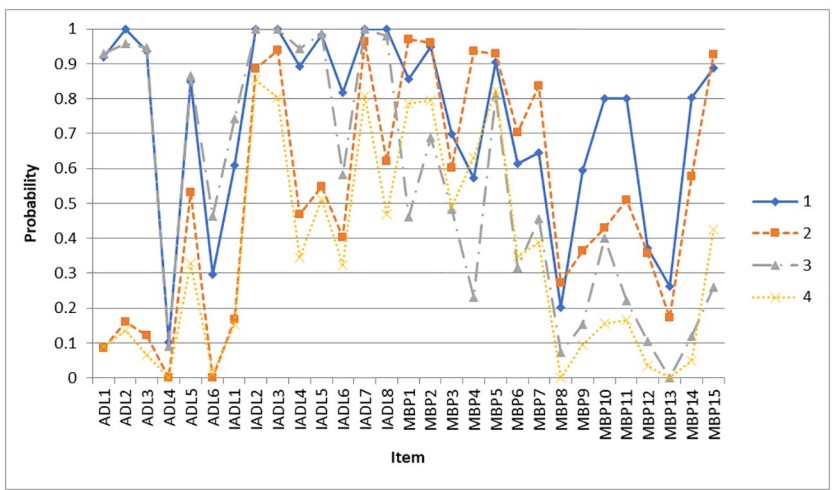

**Fig 1. Conditional probability of class based on PWD symptoms (answer = yes).**

**Table 3. Statistics for different stages of dementia in the current sample.**

| | Average diagnosis duration | No of ADL | No of IADL | No of MBP | Prevalence of potential depression among caregivers |
|---|---|---|---|---|---|
| Early stage (class 4, n = 73) | 38.2 | 0.6 | 4.2 | 5.1 | 31.5% |
| Mild stage (class 2, n = 71) | 48.4 | 0.9 | 5.0 | 9.6 | 45.1% |
| Moderate stage (class 1, n = 47) | 54.7 | 4.1 | 7.3 | 10.0 | 55.3% |
| Severe stage (class 3, n = 91) | 65.7 | 4.3 | 7.2 | 4.7 | 26.4% |

Note: ADL—Activities of Daily Living Scale; IADL—Instrumental Activities of Daily Living Scale; MBP—memory and behavior problems

**Table 4. Logistic regression results of stage of PWD on potential depression among caregivers.**

| | Odds ratio | 95% Confidence Interval | | p-value |
|---|---|---|---|---|
| Early stage (class 4, n = 73) | 0.357 | 0.153 | 0.829 | **0.017**[*] |
| Mild stage (class 2, n = 71) | 0.485 | 0.215 | 1.094 | 0.081 |
| Moderate stage (class 1, n = 47) | Ref | | | |
| Severe stage (class 3, n = 91) | 0.287 | 0.13 | 0.632 | **0.002**[*] |

Note: This regression had controlled for caregiver socio-demographic characteristics (including age, gender, ethnicity, education level, marital status, employment status, and personal monthly income) and caregiving related variables (including caregiver's relationship to the PWD, living arrangement, having a domestic helper or not, and caregiving duration);

[*]p<0.05.

The logistic regression results suggested that after controlling for confounders (i.e. all variables listed in Table 1), the prevalence of potential depression based on CES-D cutoff of 16 was still significantly different amongst caregivers who were taking care of PWD at different stages—early stage vs. moderate stage (odds ratio = 0.357, 95% CI 0.153–0.829); and severe stage vs. moderate stage (odds ratio = 0.287, 95% CI 0.130–0.632) of dementia. Please refer to Table 4 for the details.

Since the LCA suggested similar patterns of ADL and IADL across the stages—low ADL and low IADL for both early and mild stages; and high ADL and high IADL for both moderate and severe stages, to simplify the process of obtaining cut-offs values for staging, these two were added together to form a single new variable (i.e. functional dependence). As a result, the four stages derived from the LCA can be represented by a 2x2 matrix on level of functional dependence and MBP level: 1) early stage—low functional dependence & low MBP; 2) mild stage—low functional dependence & high MBP; 3) moderate stage—high functional dependence & high MBP; and 4) severe stage—high function dependence & low MBP. Since the pattern of functional dependence and MBP are inconsistent across the stages, two cut-off values were calculated based on functional dependence (i.e. ADL+IADL) and the RMBPC separately. Logistic regression with ROC analysis was conducted for stages showed high levels of functional dependence (=1 if PWD fell under moderate or severe stages; = 0 otherwise), with functional dependence being the predictor. Another logistic regression with ROC analysis was also conducted for stages showed high level of MBP (=1 if PWD fell under mild or moderate stages; = 0 otherwise), regressing on the RMPBC. The former logistic regression suggested that when the probability level fell between 0.08 and 0.56, the cut-off value of functional dependence show highest Youden's J statistic (i.e. 0.993+0.951–1 = 0.944). This converted to a range of the cut-off values for functional dependence from 8.11 to 8.95. Since functional dependence is an integer, 9 was selected as its cut-off (i.e. sum of ADL and IADL). Similarly, for level of MBP, Youden's J statistic (0.89+0.957–1 = 0.847) was highest when the probability level was between 0.28 and 0.82, which converted to a range of cut-off values between 7.01 and 7.97. As a result, 8 was selected as the cut-off for the RMBPC. In this case for this screening strategy, PWD will be allocated to early stage if their score on ADL+IADL < 9 and RMBPC < 8; mild stage if ADL+IADL < 9 and RMBPC > = 8; moderate stage if ADL+IADL > = 9 and RMBPC > = 8; and severe stage if ADL+IADL > = 9 and RMBPC < 8. The screening based on these two cut-off values achieved an overall accuracy of 90.8% as compared to the stages derived from the LCA. Refer to Table 5 for the numbers under each stage from the LCA and the screening strategy based on these two cut-off values.

**Table 5. Comparison of number of participants under each stage.**

| | | Latent class analysis staging | | | |
|---|---|---|---|---|---|
| | | **Early** | **Mild** | **Moderate** | **Severe** |
| Screening staging | Early | **69** | 8 | 0 | 0 |
| | Mild | 2 | **58** | 1 | 0 |
| | Moderate | 0 | 3 | **43** | 5 |
| | Severe | 2 | 2 | 3 | **86** |

## Discussion

The latent class analysis revealed a 4-class solution based on caregiver reported symptoms of PWD. These four classes were mutually exclusive and showed a clear trend reflecting PWD's duration of diagnosis. This supports our hypothesis that staging dementia purely based on caregiver reported PWD symptoms is a viable approach. Unlike other diseases, primary informal caregivers of PWD are the most suitable persons to report PWDs' symptoms since they are the family members or friends who are most involved in PWDs' daily care. From this point of view, this new staging mechanism derived from the LCA is reliable.

The new staging strategy classified PWD into four stages, namely early stage, mild stage, moderate stage, and severe stage. At the very beginning of the disease (early stage), PWD show very minimum dependence in ADL, IADL, and have relatively fewer issues on memory and behavior problems. This stage usually lasts for a few years until the disease progresses to the mild stage. During the mild stage, PWD have more memory and behavior problems; however, their abilities for ADL and IADL remain similar as in the early stage. During this stage, PWD might cause caregiver stress as they are physically independent but mentally dependent. Gradually, PWD move to the moderate stage in which PWD's dependence on others for their ADL and IADL increases, but their level of memory and behavior problems remains similar as at the mild stage. This stage is usually most stressful for caregivers since they need to take care of both the physical needs and the memory and behavior issues of the PWD. This could also be seen from the higher odds of potential depression identified in this group of caregivers. Lastly, PWD move to the severe stage, in which PWD are mostly bed-ridden, with high dependence on ADL and IADL, and caregivers are less troubled by PWDs' memory and behavior problems.

Other than the staging from the LCA, the current study also provides cut-offs on functional dependence (i.e., 9 for ADL + IADL) and level of memory and behavior problems (i.e., 8 for RMBPC), which enable caregivers to quickly assess which stage the PWD they are caring for belong to. This screening strategy had an overall accuracy of 90.8% among the current population. Moreover, unlike the traditional classification which usually needs to be assessed by trained healthcare professionals [20], this approach only requires caregivers' rating on PWD's dependence on the ADL and the IADL, and presence of symptoms on the RMBPC. This makes it much more operational compared to traditional clinical assessments and as a result requires much less resources if implemented in practice.

Findings from the current study have significant implications, especially in the area of tailoring interventions for caregivers. Firstly, since this new staging strategy only relies on caregivers' rating of PWD's dependence on the ADL, IADL, and symptoms on the RMBPC, it can be applied across all diagnoses of dementia. Secondly, through this new strategy, caregivers can easily figure out the stage of the PWD they are taking caring of, which provides the foundation for the group tailoring of support towards the caregiver. Given that there were successful attempts of organizing caregiving considerations based on stages of

Alzheimer's disease [44, 45], and qualitative evidence on caregiving experiences revealed similar key themes and caregiving needs despite caring for PWD with different diagnosis [7, 9, 46, 47], we believe that caregiving consideration of other types of dementia could also be organized similarly. For example, caregivers taking care of PWD in the early stage mostly need to deal with issues such as role transition and acceptance, and advice on dealing with PWDs' memory and behavior problems; while for PWD at mild stage, priority should be on tips for handling PWDs' memory and behavior problems and the emerging functional dependence. Thus, service or support providers could tailor their training and support for caregivers based on the typical needs of each specific stage the PWD belongs to. Such customization has a huge potential in e-health/m-health [17], in the sense that service providers could organize all information or resources based on stage of PWD, so that caregivers could specify which stage of dementia their relative is in and then receive the most pertinent support without an overload of information, which can lead to better outcomes for both the providers and the PWD.

## Strengths and limitations

To the best of our knowledge, this is the first study in Asia which had explored the possibility of staging dementia purely based on caregiver reported PWD symptoms. It is also the first study that aimed to provide the screening cut-offs for each stage based on the tools used to assess PWD symptoms. With these cut-offs, caregivers could easily determine the stage of the PWD they are caring for. Since this new strategy does not require any specific training, it has a huge potential to be widely and easily used. A previous study also staged dementia based on patient symptoms; however, it relied on artificial intelligence and failed to provide a practical and straightforward solution for caregivers to figure out the PWDs' stage by themselves [21]. For future studies, researchers could focus on investigating the typical challenges faced by caregivers when they are taking care of PWD at specific stages so that tailored support could be provided to caregivers depending on PWDs' stage.

The study findings should be viewed with the following limitations in mind. First of all, the study sample comprised only primary informal caregivers of PWD recruited through convenience sampling in Singapore, and they voluntarily chose to join the study. This might cause self-selection bias and might limit the applicability of the new staging strategy among caregivers and PWDs elsewhere. More studies on the generalizability of this new strategy are needed. Secondly, the data were collected through interviewer-administered questionnaire reported by caregivers, and this may have led to social desirability bias. The rating might be influenced by caregivers' own mental health status as well. Thirdly, readers should bear in mind that the diagnosis duration could be shortened by potential late diagnosis of dementia, and the deterioration of PWD's functional status could be caused by the comorbid conditions of PWD. Fourthly, we failed to collect the clinical diagnosis of the PWD, and as a result, are unable to provide it as a reference, Lastly, the LCA classified the PWD into four different classes based on patterns of caregivers reported patient symptoms; however, this classification might differ from those derived from clinical assessments. Since we failed to collect the clinical stage information, we were unable to provide such comparisons. In this case, it will be ideal for future studies to compare this staging strategy with other clinical assessment tools to cross-validate it. Nonetheless, these should not lessen the novelty, the operability, and the huge potential of this new staging strategy, and this new strategy is a good complement of existing clinical assessment tools in terms of better supporting informal dementia caregivers.

## Conclusion

The current study confirmed that caregiver reported patient symptoms could be used to classify PWD into different stages. PWD under each stage had their own symptom patterns, with the mild and moderate stages showing more memory and behavior problems and, as a result posed more stress to their caregivers. Further analyses generated cut-off values on functional dependence (i.e., ADL + IADL) and level of memory and behavior problems (i.e., RMBPC) separately. And these two cut-off values, when used together, could assign 90.8% of the PWD into the right stages derived from the LCA. The new staging strategy is a good complement to existing dementia clinical assessment tools in terms of better supporting informal caregivers.

## Acknowledgments

The authors would like to thank all the participants for their time and efforts in the study.

## Author Contributions

**Conceptualization:** Qi Yuan, Mythily Subramaniam.

**Data curation:** Qi Yuan, Tee Hng Tan.

**Formal analysis:** Qi Yuan, Edimansyah Abdin.

**Funding acquisition:** Qi Yuan, Siow Ann Chong.

**Investigation:** Qi Yuan, Tee Hng Tan, Peizhi Wang, Fiona Devi, Hui Lin Ong.

**Methodology:** Qi Yuan, Edimansyah Abdin.

**Project administration:** Qi Yuan.

**Resources:** Magadi Harish, Richard Goveas, Li Ling Ng, Siow Ann Chong, Mythily Subramaniam.

**Software:** Qi Yuan.

**Supervision:** Siow Ann Chong, Mythily Subramaniam.

**Visualization:** Qi Yuan.

**Writing – original draft:** Qi Yuan.

**Writing – review & editing:** Tee Hng Tan, Peizhi Wang, Fiona Devi, Hui Lin Ong, Edimansyah Abdin, Magadi Harish, Richard Goveas, Li Ling Ng, Siow Ann Chong, Mythily Subramaniam.

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
