## [Decision Letter · Decision Letter 0]

29 Nov 2019

PONE-D-19-30761

Staging Dementia based on Caregiver Reported Patient Symptoms: Implications from a Latent Class Analysis

PLOS ONE

Dear Dr Yuan,

Thank you for submitting your manuscript to PLOS ONE. After careful consideration, we feel that it has merit but does not fully meet PLOS ONE’s publication criteria as it currently stands. Therefore, we invite you to submit a revised version of the manuscript that addresses the points raised during the review process.

While both reviewers saw merit in the manuscript, a number of issues were raised.  Of note, the skewed nature of the sample population is not thoroughly discussed. The remaining queries from the reviewers are clearly expressed.

We would appreciate receiving your revised manuscript by Jan 13 2020 11:59PM. To enhance the reproducibility of your results, we recommend that if applicable you deposit your laboratory protocols in protocols.io, where a protocol can be assigned its own identifier (DOI) such that it can be cited independently in the future. For instructions see: http://journals.plos.org/plosone/s/submission-guidelines#loc-laboratory-protocols

We look forward to receiving your revised manuscript.

Kind regards,

César Leal-Costa, Ph. D

Academic Editor

PLOS ONE

Journal Requirements:

2. Please refer to any sample size calculations performed prior to participant recruitment. If these were not performed please justify the reasons. Please refer to our statistical reporting guidelines for assistance (https://journals.plos.org/plosone/s/submission-guidelines.#loc-statistical-reporting).

Additional Editor Comments (if provided):

Reviewers' comments:

Reviewer's Responses to Questions

**Comments to the Author**

1. Is the manuscript technically sound, and do the data support the conclusions?

Reviewer #1: Partly

Reviewer #2: Yes

2. Has the statistical analysis been performed appropriately and rigorously? 

Reviewer #1: Yes

Reviewer #2: I Don't Know

3. Have the authors made all data underlying the findings in their manuscript fully available?

Reviewer #1: No

Reviewer #2: No

4. Is the manuscript presented in an intelligible fashion and written in standard English?

Reviewer #1: Yes

Reviewer #2: Yes

5. Review Comments to the Author

Reviewer #1: The manuscript reports results of staging Dementia based on Caregiver Reported Patient Symptoms: implications from a Latent Class Analysis.

Generally, it aims to stage dementia based on caregiver reported symptoms of persons with dementia. Optimal cut-offs for the assessment tools used were also calculated. The paper is an interesting study that evaluates several specific questions. However, in my opinion there are several aspects should be revised to improve the explanatory power of the manuscript as noted below.

Important:

*** Do not use more than 5 abbreviations in the manuscript. After reducing the number of abbreviations, create a table that includes the abbreviations used.

*** Materials and Methods, Results and Discussion: to facilitate reading I recommend to the authors include a paragraph for each objective.

Abstract

Results: Include the main results and p-values if there was significance.

Conclusion: Delete last sentence and include it in the last part of Discusion.

Materials and Methods

a) Participants and Procedures. To create a paragraph entitled: Ethics approval and consent to participate (or similar). I recommend this text:

The protocol was approved by xxxxxx (city, country; with reference number xxxxxxxxxx). The study was developed in accordance with the precepts of the Declaration of Helsinki. All patients signed the corresponding written informed consent prior to their enrolment and appropriate measures were taken to ensure data privacy.

b) Measurements

1. Lines 120-122: It is very important to justify why were used 2 scales from 1963 and 1969 respectively to know functional status, memory and behavior problems. Please, explain the reasons why more recent scales were not used.

2. Line 132: To include reference for the checklist RMBPC.

3. Lines 139-140: To explain the reason to use the 20-item Centre for Epidemiological Study Scale from 1977. Please, explain the reason why more recent scale was not used.

4. Are the used scales validated in Singapur?

c) Data analysis

Modify this section: One paragraph for every step performed.

Results

Table 1:

a) No simple size calculations are referred to in this manuscript. Besides, the skewed nature of the simple populaton is not throughly discussed.

b) To diference between 1) mean and frequency 2) SD and percentage.

c) Include a column with p-values.

d) At the table botton include the meaning of SGD and PWD.

e) Tables 2, 3 and 5 include: a) p-values and 2) the meaning of the abreviations.

Discussion

a) First paragraph: clearly include which are the main findings.

b) I recommend a paragraph for every objective and compare the main results with previous studies.

c) Include a paragraph that includes the main strengths and weaknesses of the study.

Reviewer #2: Congratulations on the excellent work. Visiting the important role of informal caregivers is a challenge. I think your study helps to empower the figure of the caregiver of people with dementia.

However, some aspects can be improved when reporting your work in the manuscript.

* Abstract:

I recommend deleting the following text from the "objective" section: "Optimal cut-offs for the assessment tools used were also calculated. “The writing of a research objective should not include a sentence about a result or method that has been performed, but what is intended to be discovered with the study.

* The introduction section is well structured, with a good justification for the study and an adequate background.

* I understand that no probabilistic sampling has been performed and this should be explained both in the method section and in the limitations of the study in the discussion section.

* More specific information about the ethical issues of the study would be necessary. The ethical aspects are not adequately explained and are summarized in that it was approved by a committee.

* The results are adequately discussed in the manuscript. The limitations are exposed and that is a strength in scientific research. Congratulations

* The possible future lines of research have already been set out in the discussion section. It is not necessary to repeat them in the conclusion section, so you would delete that text from one section or the other.

6. PLOS authors have the option to publish the peer review history of their article (what does this mean?). If published, this will include your full peer review and any attached files.

Reviewer #1: No

Reviewer #2: No

---

## [Author Response · Author response to Decision Letter 0]

6 Dec 2019

6th Dec 2019

MS ID: PONE-D-19-30761

MS title: Staging Dementia based on Caregiver Reported Patient Symptoms: Implications from a Latent Class Analysis

Dear reviewers,

Thank you very much for the review and also for the valuable comments and suggestions. Our point-by-point responses are provided below and in bold for easy reference, and highlighted with track changes in the revised manuscript.

We hope that these changes are satisfactory. Thank you very much again in advance!

Sincerely,

Corresponding author

 

Journal Requirements:

1. Please ensure that your manuscript meets PLOS ONE's style requirements, including those for file naming. The PLOS ONE style templates can be found at https://imsva91-ctp.trendmicro.com:443/wis/clicktime/v1/query?url=http%3a%2f%2fwww.journals.plos.org%2fplosone%2fs%2ffile%3fid%3dwjVg%2fPLOSOne%5fformatting%5fsample%5fmain%5fbody.pdf&umid=4913B737-987A-2E05-A0D2-C3CB71FD25E2&auth=6e3fe59570831a389716849e93b5d483c90c3fe4-02f1250312cbd8cc3126c5bcb57b48446f45e8ae and https://imsva91-ctp.trendmicro.com:443/wis/clicktime/v1/query?url=http%3a%2f%2fwww.journals.plos.org%2fplosone%2fs%2ffile%3fid%3dba62%2fPLOSOne%5fformatting%5fsample%5ftitle%5fauthors%5faffiliations.pdf&umid=4913B737-987A-2E05-A0D2-C3CB71FD25E2&auth=6e3fe59570831a389716849e93b5d483c90c3fe4-cf0dcc3759a9e88581514f7555151ea3ba18bb04

RESPONSE: The manuscript is formatted following PLOS ONE style template.

2. Please refer to any sample size calculations performed prior to participant recruitment. If these were not performed please justify the reasons. Please refer to our statistical reporting guidelines for assistance (https://imsva91-ctp.trendmicro.com:443/wis/clicktime/v1/query?url=https%3a%2f%2fjournals.plos.org%2fplosone%2fs%2fsubmission%2dguidelines.%23loc%2dstatistical%2dreporting&umid=4913B737-987A-2E05-A0D2-C3CB71FD25E2&auth=6e3fe59570831a389716849e93b5d483c90c3fe4-bc31dbd4e458bf3b4e0e8dc40582663843766e1d).

RESPONSE: We did calculate the sample size prior to initiating the study. However, since the calculation was based on the main project outcome (i.e. prevalence of depression), the current study is an exploratory analysis based on the sample collected, which is different from the main project outcome. In this case, we thought it would be misleading if we include the sample size calculation (based on main project outcome) in the manuscript. 

3. We note that you have indicated that data from this study are available upon request. PLOS only allows data to be available upon request if there are legal or ethical restrictions on sharing data publicly. For information on unacceptable data access restrictions, please see https://imsva91-ctp.trendmicro.com:443/wis/clicktime/v1/query?url=http%3a%2f%2fjournals.plos.org%2fplosone%2fs%2fdata%2davailability%23loc%2dunacceptable%2ddata%2daccess%2drestrictions&umid=4913B737-987A-2E05-A0D2-C3CB71FD25E2&auth=6e3fe59570831a389716849e93b5d483c90c3fe4-60195c19acf64d4762b069957f2ef692d6956ca7.

b) If there are no restrictions, please upload the minimal anonymized data set necessary to replicate your study findings as either Supporting Information files or to a stable, public repository and provide us with the relevant URLs, DOIs, or accession numbers. Please see https://imsva91-ctp.trendmicro.com:443/wis/clicktime/v1/query?url=http%3a%2f%2fwww.bmj.com%2fcontent%2f340%2fbmj.c181.long&umid=4913B737-987A-2E05-A0D2-C3CB71FD25E2&auth=6e3fe59570831a389716849e93b5d483c90c3fe4-5482821ba37a53710fcbddbac167cae3ed426272 for guidelines on how to de-identify and prepare clinical data for publication. For a list of acceptable repositories, please see https://imsva91-ctp.trendmicro.com:443/wis/clicktime/v1/query?url=http%3a%2f%2fjournals.plos.org%2fplosone%2fs%2fdata%2davailability%23loc%2drecommended%2drepositories&umid=4913B737-987A-2E05-A0D2-C3CB71FD25E2&auth=6e3fe59570831a389716849e93b5d483c90c3fe4-1305d747dc89a01960ad75d3185e772a533fd7de.

RESPONSE: The authors' funding agency or government law only permits sharing of human participant data with researchers with whom they have a written agreement. The restrictions have been imposed by our Institutional Review Board (IRB) and Institutional Committee (NHG Domain Specific Review Board and IMH Clinical Research Committee). Our IRB guidelines suggest that a Research Collaboration Agreement (RCA) be signed with collaborating parties. However, data sharing with clear research purposes are available upon request to the senior author. The contact details are included in the cover letter to the editor.

Additional Editor Comments (if provided):

Reviewers' comments:

Reviewer's Responses to Questions

 

Comments to the Author

1. Is the manuscript technically sound, and do the data support the conclusions?

Reviewer #1: Partly

Reviewer #2: Yes

2. Has the statistical analysis been performed appropriately and rigorously? 

Reviewer #1: Yes

Reviewer #2: I Don't Know ________________________________________

3. Have the authors made all data underlying the findings in their manuscript fully available?

Reviewer #1: No

Reviewer #2: No________________________________________

4. Is the manuscript presented in an intelligible fashion and written in standard English?

Reviewer #1: Yes

Reviewer #2: Yes________________________________________

5. Review Comments to the Author

 

Reviewer #1: The manuscript reports results of staging Dementia based on Caregiver Reported Patient Symptoms: implications from a Latent Class Analysis.

Generally, it aims to stage dementia based on caregiver reported symptoms of persons with dementia. Optimal cut-offs for the assessment tools used were also calculated. The paper is an interesting study that evaluates several specific questions. However, in my opinion there are several aspects should be revised to improve the explanatory power of the manuscript as noted below.

RESPONSE: Thank you very much!

Important:

*** Do not use more than 5 abbreviations in the manuscript. After reducing the number of abbreviations, create a table that includes the abbreviations used.

RESPONSE: We would like to thank the reviewer for this comment. According to the 6th APA guideline, it suggested that abbreviations should be used when the term appears at least three times in the manuscript, and there are no hard lines regarding how many abbreviations are too many. We have revised the manuscript following this guideline, keeping those abbreviations that appears at least three times. And as the reviewer suggested, we have created a list of abbreviations in the end of the manuscript. Please refer to the revised manuscript for more details.

*** Materials and Methods, Results and Discussion: to facilitate reading I recommend to the authors include a paragraph for each objective.

RESPONSE: We are sorry that we are not quite sure about what exactly the reviewer is suggesting. In fact, the current manuscript is organized the same way as all other published articles in the journal. May we request that this point be further clarified with an example from any other articles?

Abstract

Results: Include the main results and p-values if there was significance.

RESPONSE: For the latent class analysis and the accuracy statistics in the results section of the abstract, both didn’t involve any statistical comparisons, thus there were no relevant p-values. We had made some necessary changes to the results section in the revisions to make it clearer.

Conclusion: Delete last sentence and include it in the last part of Discusion.

RESPONSE: The manuscript has been revised accordingly.

Materials and Methods

a) Participants and Procedures. To create a paragraph entitled: Ethics approval and consent to participate (or similar). I recommend this text:

The protocol was approved by xxxxxx (city, country; with reference number xxxxxxxxxx). The study was developed in accordance with the precepts of the Declaration of Helsinki. All patients signed the corresponding written informed consent prior to their enrolment and appropriate measures were taken to ensure data privacy.

RESPONSE: In fact, we did have a separate paragraph on the study ethics (i.e. the last paragraph of the ‘Participants and Procedures’ section. The ethics statement has been revised accordingly as suggested by the reviewer.

b) Measurements

1. Lines 120-122: It is very important to justify why were used 2 scales from 1963 and 1969 respectively to know functional status, memory and behavior problems. Please, explain the reasons why more recent scales were not used.

RESPONSE: We would like to apologize for not clarifying this in the original manuscript. There are several reasons: first of all, these two scales are both very classical, and there are plenty of articles demonstrating their reliability. Secondly, it has been validated in Singapore (ADL [1, 2] and IADL [3]), and also been used among caregivers in Singapore before, showed good internal reliability [4]. Lastly, it is short and easily administered among dementia caregivers who usually have an average age of 50 years and more.

2. Line 132: To include reference for the checklist RMBPC.

RESPONSE: In fact, the reference of RMBPC was included in the manuscript – in the next sentence while describing the checklist. To avoid confusion, we have added in the citation in line 132 (now line 135) as suggested by the reviewer.

3. Lines 139-140: To explain the reason to use the 20-item Centre for Epidemiological Study Scale from 1977. Please, explain the reason why more recent scale was not used.

RESPONSE: There are a few reasons: 1) according the American Psychological Association [5], the CES-D provides cut-off scores that aid to identify individuals at risk for clinical depression, with good sensitivity and specificity and high internal consistency [6], and it is sensitive to difference between caregivers and non-caregivers [7], and it can be used across wide age ranges [6]; 2) it has been validated in Singapore before [8]; 3) a previous Singapore study suggested that it has good validity and utility in detecting depression among family caregivers of PWD [9].

4. Are the used scales validated in Singapur?

RESPONSE: Please refer to our responses to comments (answers to b) measurement – point 1 and point 3) 

c) Data analysis

Modify this section: One paragraph for every step performed.

RESPOSE: To make it clearer to the audience, we use numbered list to present the four steps of the analyses. Please refer to the revised manuscript for the details.

Results

Table 1:

a) No simple size calculations are referred to in this manuscript. Besides, the skewed nature of the simple populaton is not throughly discussed.

b) To diference between 1) mean and frequency 2) SD and percentage.

c) Include a column with p-values.

d) At the table botton include the meaning of SGD and PWD.

e) Tables 2, 3 and 5 include: a) p-values and 2) the meaning of the abreviations.

RESPONSE: a) We did calculate the sample size prior to initiating the study. However, since the calculation was based on the main project outcome (i.e. prevalence of depression), the current study is an exploratory analysis based on the sample collected, which is different from the main project outcome. In this case, we thought it would be misleading if we include the sample size calculation (based on main project outcome) in the manuscript. 

Our study sample is informal dementia caregivers, which in nature would differ from the general population. However, this doesn’t necessarily suggest that our sample is skewed. In fact, we compared our sample characteristics with the sample characteristics of dementia caregivers from a previous Singapore Dementia Caregiver Profile study (SDCP) [10], and we found very similar statistics. For example, average age: current study vs. SDCP – 55.7 (11.8) vs. 52.6 (11.1); female proportion: current study vs. SDCP – 75.2% vs. 73.6; relationship with PWD: current study vs. SDCP – 15.3% vs. 18% (spouse), 72.3% vs. 69.2% (children), and 12.4% vs. 12.8% (others); Living with PWD: current study vs. SDCP – 70.2% vs. 79.2%; and having a domestic maid: current study vs. SDCP – 57.1% vs. 52.8%. But we also understand the self-selection bias might affect the generalizability of the study findings, thus this limitation was added.

b) Table 1 was re-formatted accordingly.

c) Table 1 includes only the descriptive statistics, p-value is not applicable here.

d) The meaning of SGD and PWD were added as suggested.

e) Table 2 is the latent class analysis results, table 3 and table 5 both included descriptive statistics, all these three tables don’t involve p-values. 

As suggested by the reviewer, we have added in the meaning of all abbreviations as the notes of the tables.

Discussion

a) First paragraph: clearly include which are the main findings.

b) I recommend a paragraph for every objective and compare the main results with previous studies.

c) Include a paragraph that includes the main strengths and weaknesses of the study.

RESPONSE: a) Our main finding is that staging dementia based on caregivers reported is a viable approach which had been included in the first paragraph of the discussion section. However, we do feel the original first paragraph included too many details; thus, we have cut it into two paragraphs so that the new first paragraph is focused on the main study findings, and the following paragraph is focused on the details. 

b) After the above revisions, the structure of the discussion section follows exactly what the reviewer suggested. In fact, this is the first study which uses such analytical strategies among informal dementia caregivers. There was one study which used artificial intelligence to stage dementia based on caregiver reported patient symptoms, although we had included it in the original manuscript, since the two use quite different methodologies, we felt it was not appropriate for us to directly compare them. Other than that, we cannot identify any other similar previous studies for us to compare. 

c) We are sorry for the confusion. In fact, the last two paragraphs of the discussion section were the study strengths and limitations. In the revision, we have added a section named ‘Strengths and Limitations’ to avoid confusion.

 

Reviewer #2: Congratulations on the excellent work. Visiting the important role of informal caregivers is a challenge. I think your study helps to empower the figure of the caregiver of people with dementia.

However, some aspects can be improved when reporting your work in the manuscript.

RESPONSE: Thanks!

* Abstract:

I recommend deleting the following text from the "objective" section: "Optimal cut-offs for the assessment tools used were also calculated. “The writing of a research objective should not include a sentence about a result or method that has been performed, but what is intended to be discovered with the study.

RESPONSE: We would like to thank the reviewer for this comment, the abstract was revised accordingly.

* The introduction section is well structured, with a good justification for the study and an adequate background.

RESPONSE: Thanks!

* I understand that no probabilistic sampling has been performed and this should be explained both in the method section and in the limitations of the study in the discussion section.

RESPONSE: We are sorry for the negligence. We have added in some descriptions on the non-probabilistic sampling in the methods and study limitations sections, please refer to the revised manuscript for more details. 

* More specific information about the ethical issues of the study would be necessary. The ethical aspects are not adequately explained and are summarized in that it was approved by a committee.

RESPONSE: We appreciate the reviewer for this comment. More details regarding the ethics were added as suggested.

* The results are adequately discussed in the manuscript. The limitations are exposed and that is a strength in scientific research. Congratulations

RESPONSE: Thanks!

* The possible future lines of research have already been set out in the discussion section. It is not necessary to repeat them in the conclusion section, so you would delete that text from one section or the other.

RESPONSE: As suggested by the reviewer, the future lines of research in the ‘Conclusion’ section was removed to avoid duplicity. 

6. PLOS authors have the option to publish the peer review history of their article (what does this mean?). If published, this will include your full peer review and any attached files.

Do you want your identity to be public for this peer review? For information about this choice, including consent withdrawal, please see our Privacy Policy.

Reviewer #1: No

Reviewer #2: No

References

1. Sien NY, Jung HJTSFP. Assessment of the six activities of daily living in adults. 2014;40(4):26-36.

2. Xie F, Li S-C, Roos EM, Fong K-Y, Lo N-N, Yeo S-J, et al. Cross-cultural adaptation and validation of Singapore English and Chinese versions of the Knee injury and Osteoarthritis Outcome Score (KOOS) in Asians with knee osteoarthritis in Singapore. 2006;14(11):1098-103.

3. Ng T-P, Niti M, Chiam P-C, Kua E-HJTJoGSABS, Sciences M. Physical and cognitive domains of the instrumental activities of daily living: validation in a multiethnic population of Asian older adults. 2006;61(7):726-35.

4. Østbye T, Malhotra R, Malhotra C, Arambepola C, Chan AJJoGSBPS, Sciences S. Does support from foreign domestic workers decrease the negative impact of informal caregiving? Results from Singapore survey on informal caregiving. 2013;68(4):609-21.

5. The American Psycholiglcal Association. Center for Epidemiological Studies-Depression [cited 2019 2 Dec]. Available from: https://www.apa.org/pi/about/publications/caregivers/practice-settings/assessment/tools/depression-scale.

6. Lewinsohn PM, Seeley JR, Roberts RE, Allen NBJP, aging. Center for Epidemiologic Studies Depression Scale (CES-D) as a screening instrument for depression among community-residing older adults. 1997;12(2):277.

7. Pinquart M, Sörensen SJP, aging. Differences between caregivers and noncaregivers in psychological health and physical health: a meta-analysis. 2003;18(2):250.

8. Stahl D, Sum CF, Lum SS, Liow PH, Chan YH, Verma S, et al. Screening for depressive symptoms: validation of the center for epidemiologic studies depression scale (CES-D) in a multiethnic group of patients with diabetes in Singapore. 2008;31(6):1118-9.

9. Ying J, Yap P, Gandhi M, Liew TMJD, disorders gc. Validity and utility of the center for epidemiological studies depression scale for detecting depression in family caregivers of persons with dementia. 2019;47(4-6):323-34.

10. Tew CW, Tan LF, Luo N, Ng WY, Yap PJD, Disorders GC. Why family caregivers choose to institutionalize a loved one with dementia: a Singapore perspective. 2010;30(6):509-16.

---

## [Decision Letter · Decision Letter 1]

24 Dec 2019

PONE-D-19-30761R1

Staging Dementia based on Caregiver Reported Patient Symptoms: Implications from a Latent Class Analysis

PLOS ONE

Dear Dr Yuan,

Thank you for submitting your manuscript to PLOS ONE. After careful consideration, we feel that it has merit but does not fully meet PLOS ONE’s publication criteria as it currently stands. Therefore, we invite you to submit a revised version of the manuscript that addresses the points raised during the review process.

The revised manuscript is much improved and has addressed many of the concerns of the reviewers. However, some issues remain outstanding and warrant attention. Notably, Reviewer 1 raises the query about the table 1.

We would appreciate receiving your revised manuscript by Feb 07 2020 11:59PM. To enhance the reproducibility of your results, we recommend that if applicable you deposit your laboratory protocols in protocols.io, where a protocol can be assigned its own identifier (DOI) such that it can be cited independently in the future. For instructions see: http://journals.plos.org/plosone/s/submission-guidelines#loc-laboratory-protocols

We look forward to receiving your revised manuscript.

Kind regards,

César Leal-Costa, Ph. D

Academic Editor

PLOS ONE

Reviewers' comments:

Reviewer's Responses to Questions

**Comments to the Author**

1. If the authors have adequately addressed your comments raised in a previous round of review and you feel that this manuscript is now acceptable for publication, you may indicate that here to bypass the “Comments to the Author” section, enter your conflict of interest statement in the “Confidential to Editor” section, and submit your "Accept" recommendation.

Reviewer #1: All comments have been addressed

Reviewer #2: (No Response)

2. Is the manuscript technically sound, and do the data support the conclusions?

Reviewer #1: Yes

Reviewer #2: (No Response)

3. Has the statistical analysis been performed appropriately and rigorously? 

Reviewer #1: Yes

Reviewer #2: (No Response)

4. Have the authors made all data underlying the findings in their manuscript fully available?

Reviewer #1: Yes

Reviewer #2: (No Response)

5. Is the manuscript presented in an intelligible fashion and written in standard English?

Reviewer #1: Yes

Reviewer #2: (No Response)

6. Review Comments to the Author

Reviewer #1: First of all I would like to thank the authors of this manuscript for having addressed the issues raised previously (I think the article has been greatly improved, in my opinion).

I just have to make two points:

1) Table 1: in order that the results can be compared to other studies I recommend converting the currency "Singapore dollars" to "Euros or American dollars".

2) Strengths and limitations: delete paragraph 1 and put it at the end of the Discussion.

Thank you,

Reviewer #2: (No Response)

7. PLOS authors have the option to publish the peer review history of their article (what does this mean?). If published, this will include your full peer review and any attached files.

Reviewer #1: No

Reviewer #2: No

---

## [Author Response · Author response to Decision Letter 1]

26 Dec 2019

27th Dec 2019

MS ID: PONE-D-19-30761R1

MS title: Staging Dementia based on Caregiver Reported Patient Symptoms: Implications from a Latent Class Analysis

Dear reviewers,

Thank you very much for the review and also for the valuable comments and suggestions. Our point-by-point responses are provided below and in bold for easy reference, and highlighted with track changes in the revised manuscript.

We hope that these changes are satisfactory. Thank you very much again in advance!

Sincerely,

Corresponding author

 

Comments to the Author

1. If the authors have adequately addressed your comments raised in a previous round of review and you feel that this manuscript is now acceptable for publication, you may indicate that here to bypass the “Comments to the Author” section, enter your conflict of interest statement in the “Confidential to Editor” section, and submit your "Accept" recommendation.

Reviewer #1: All comments have been addressed

Reviewer #2: (No Response)

RESPONSE: Thanks! 

2. Is the manuscript technically sound, and do the data support the conclusions?

Reviewer #1: Yes

Reviewer #2: (No Response)

3. Has the statistical analysis been performed appropriately and rigorously? 

Reviewer #1: Yes

Reviewer #2: (No Response)

4. Have the authors made all data underlying the findings in their manuscript fully available?

Reviewer #1: Yes

Reviewer #2: (No Response)

5. Is the manuscript presented in an intelligible fashion and written in standard English?

Reviewer #1: Yes

Reviewer #2: (No Response)

6. Review Comments to the Author

Reviewer #1: First of all I would like to thank the authors of this manuscript for having addressed the issues raised previously (I think the article has been greatly improved, in my opinion).

RESPOSE: Thank you very much!

I just have to make two points:

1) Table 1: in order that the results can be compared to other studies I recommend converting the currency "Singapore dollars" to "Euros or American dollars".

RESPONSE: We would like to thank the reviewer for this comment. In the revision, we have added in the income level as converted into US dollars to enable easier comparisons with international studies.

2) Strengths and limitations: delete paragraph 1 and put it at the end of the Discussion.

RESPONSE: We would like to clarify – in fact that section is about strengths and limitations. The first paragraph is on strengths and the second paragraph is on limitations. In this case, it makes more sense for us to keep the current structure. 

Thank you,

Reviewer #2: (No Response)

7. PLOS authors have the option to publish the peer review history of their article (what does this mean?). If published, this will include your full peer review and any attached files.

Do you want your identity to be public for this peer review? For information about this choice, including consent withdrawal, please see our Privacy Policy.

Reviewer #1: No

Reviewer #2: No

---

## [Editor Report · Decision Letter 2]

2 Jan 2020

Staging Dementia based on Caregiver Reported Patient Symptoms: Implications from a Latent Class Analysis

PONE-D-19-30761R2

Dear Dr. Yuan,

We are pleased to inform you that your manuscript has been judged scientifically suitable for publication and will be formally accepted for publication once it complies with all outstanding technical requirements.

With kind regards,

César Leal-Costa, Ph. D

Academic Editor

PLOS ONE
---

## [Editor Report · Acceptance letter]

6 Jan 2020

PONE-D-19-30761R2 

Staging Dementia based on Caregiver Reported Patient Symptoms: Implications from a Latent Class Analysis 

Dear Dr. Yuan:

I am pleased to inform you that your manuscript has been deemed suitable for publication in PLOS ONE. Congratulations! Your manuscript is now with our production department. 

With kind regards,

on behalf of

Dr. César Leal-Costa 

Academic Editor

PLOS ONE